# Compulsory Heterosexuality in Indonesia: A Literary Exploration of the Work of Ayu Utami

**Santi Widianti**

Bandung Institute of Governance Studies, Bandung 40271, Indonesia; santi.widianti@gmail.com

**Abstract:** The fall of Suharto from the presidency in the Reformation of 1998 created space for greater freedom of expression, including for women, in the world's largest Muslim-majority nation, Indonesia. Ayu Utami is an Indonesian writer whose first novel, *Saman*, was published at the time of the Indonesian Reformation to critical and commercial acclaim. Other women writers followed her lead, expressing their work on women and sexuality. Through her writing, Utami challenges a patriarchal culture which continues to marginalise women. This paper focuses on the two of Utami's literary works, *Si Parasit Lajang: Seks, Sketsa dan Cerita* (*The Single Parasite: Sex, Sketches and Stories*) *(2003)* and *Pengakuan Eks Parasit Lajang* (*Confessions of a Former Single Parasite*) *(2013)*. These two books have parallel themes, representing Utami's challenges to dominant discourses on women and marriage. In the first book, Utami shows that women face discrimination if they remain single. Her political stance to remain unmarried is a way to show that women can choose alternative ways of life, rather than submit to the valorised option: to get married. Utami's shift of position, as elaborated in the second book through the story of her autobiographical character A, who is a Catholic, has to do with a developing tendency of the Islamic conservatism in Indonesia to silence the expression of minority groups, including minority religious communities. This paper argues that to understand Utami's shift in position on marriage, we must understand the ways in which her position as a member of a minority and, sometimes endangered, religious community shapes her position.

**Keywords:** Indonesia; women; freedom of expression; marriage; religion

## 1. Introduction

> *This society glorifies marriage too much.*
>
> *(Ayu Utami, the Jakarta Post, Junaidi 2005)*

Indonesia's reformation in 1998 has brought changes from the authoritarian regime of Suharto to an emerging democracy. Reformation was a period that was deemed momentous, as Indonesian people could express themselves after 32 years of silence. The downfall of Suharto brought with it a greater freedom of speech and expression. Not limited to the political sphere, the liberation of expression also includes more freedom of expressions in the arts. Indonesian women, who were confined to enforced sex roles as wives and mothers during the Suharto's New Order, have increasingly participated in public debates about issues relevant to women. They can redefine female identities and sexualities through various modes of creative arts. Many are optimistic that the Reformation will provide space for advocating women's rights in a free and democratic atmosphere. However, it is the Islamic conservatism that pushes its way to homogenise the diversity and pluralism in Indonesia.

Diversity and plurality are basic characteristics of Indonesian society. Although Indonesia is the largest Muslim country in the world where the majority of the population is Muslim, it mostly holds a moderate view towards their religion. In addition to Islam, there is Catholicism, Protestantism, Hinduism, Buddhism, Confucianism, and local system beliefs. Historically, Islam in Indonesia has mixed with elements of local beliefs and customs

during the period of conversion. Clifford Geertz (Geertz 1976) famously categorises Islam in Indonesia into two camps, called santri and abangan (Geertz 1976; Smith-Hefner 2019). The former is traditionalist Islam leaning toward the Middle Eastern Islam, while the latter is secular. Santri Islam is also divided again into two groupings: modernist and traditionalist. Islamic groups had a significant contribution in assisting Suharto in 1965 by joining military forces to destroy the communist party in 1965. Yet, during Suharto's New Order, Islamic groups were kept at a distance for fearing that they would threaten his position of power (Smith-Hefner 2019, p. 31; Davies 2019, p. 1068).

In stabilising the nation and maintaining social order, Suharto's New Order regulated sexuality within the marriage and constructed the nation as a family with the principle of azas kekeluargaan, or family principle (Suryakusuma 1996; Davies 2019). Davies (2019) asserts that in this family principle, "heterosexuality is the only sexuality". Furthermore, cultural customs and religion in Indonesia also emphasise values that female sexuality is only considered legitimate in the marriage (Bennett 2005a). Female sexuality is equated with reproduction to prepare a new generation of the nation. In her analysis of the linkages between the state and sexuality during Suharto's New Order, Suryakusuma (1996) argues that the New Order created state ibuism and housewification, which confined women to the domestic sphere. Jacubowski (2008, p. 90) points out that the New Order state promoted both the idealisation of women as mothers and the involvement of women in the process of development. As a result, there was an increase in women's participation in education and employment. Through the Suharto's New Order educational programs and policies (Smith-Hefner 2019), the new middle class emerged.

In the New Order, it was the state which controlled sexuality, whereas in the aftermath of the Reformation it is the conservative Islamic groups that actively enforce morality based on their own version of morality. They persecute any sexual relationships outside of heterosexual marriage, including premarital sex, and make efforts to criminalise it, while the liberal Muslims put forward principles of human rights, pluralism, and social equality regarding sexuality and morality issues (Brenner 2011). Over the years, the conservatives have grown increasingly popular among Indonesian youths. The advance of new technology and media is embraced by the Islamic conservatives to promote their ideas to wider audiences, including a new generation of middle class Muslims. This innovative use of new media particularly appeals to the younger generation. Established moderate Muslim authorities are overshadowed by new conservative preachers since they still rely on old ways of preaching. Gradually, conservative Islam is succeeding in creating their own authoritative figures and forming alliances with political elites who have vested interests in gaining political support or maintaining their position from the mass of conservative Muslims (Arifianto 2019).

It is against this backdrop that Indonesian writer Ayu Utami appeared with her work. Utami's essays and novels operate as critiques of gendered and religious oppression. In 1998 Utami managed to publish her first novel, *Saman*, during the social and political changes in Indonesia. *Saman* was published around the time when the Indonesia's reformation was about to happen—less than a fortnight before President Suharto who, after more than 30 years of power in Indonesia, was made to resign (Hatley 2007, Inside Indonesia, n.p.; Partogi 2018, the Jakarta Post). The novel was considered controversial due to its frank content on themes of sex and sexuality. *Saman* was praised for its rich language and unusual theme (Hatley 1999, p. 449). Detractors viewed *Saman* as "too vulgar, too brave, too much" (Mahditama 2012, n.p.).

In the history of literary texts in Indonesia, this was the first time a woman had written openly about sexual matters, a feature of a new atmosphere of freedom in Indonesia. Since the release of *Saman* in 1998, Utami has written several other books. In 2001, Utami published *Larung*, a sequel to *Saman*. Two years later, she published a compilation of essays, *Si Parasit Lajang: Seks, Sketsa dan Cerita* (*The Single Parasite: Sex, Sketches and Stories*), in 2003. In this book, Utami declared that she would never marry as a challenge to patriarchal culture in a society that glorifies the institution of marriage (Utami 2003, p. 169). The book

was a best seller and has since been reprinted many times. Her fourth novel, *Bilangan Fu* (*Fu Number*), was published in 2008, charting the emergence of radical religious groups, which pose a threat to the diversity of Indonesian society. In February 2013, Utami launched her new book, *Pengakuan Eks Parasit Lajang* (*Confessions of a Former Single Parasite*). This latest book is her autobiography in the form of a novel, told through the character A, a member of Indonesia's Catholic community, and explains her reasons for finally marrying a man after proclaiming years before that marriage was out of the question. This paper analyses Utami's shift in position from remaining single to eventually getting married, as outlined in: *The Single Parasite: Sex, Sketches and Stories* and *Confessions of a Former Single Parasite*. In 2003, Utami had declared that she would never marry in *The Single Parasite*. The institution of marriage has always been considered important for women in Indonesia. It is a crucial stage when a woman is finally viewed as an adult. Before she marries, a single woman is not considered an adult in Indonesia. Therefore, Utami's political stance against marriage was a challenge to prove that women's decision to remain single, as a matter of choice, is also an adult decision. In 2011, contradicting her prior declaration, Ayu Utami did get married, an attitudinal change that shocked her fans (Utami 2013, p. 2).

## 2. Single Women and Marriage

In Indonesia, there are terms such as *perawan tua* (old virgin), which has the same meaning as terms "old maid" or "spinster" in Western countries, and *tidak laku* (un-saleable) (Utomo et al. 2007, p. 4; Smith-Hefner 2005, p. 444) to refer to single women who are not married. Single women have a greater burden than their male counterparts, as women are socialised to become wives and mothers as part of an adult life stage. Situmorang (2007, p. 288) points out that singlehood is seen as a form of social failure for most people in Indonesia, but particularly for women. Although some men also remain single, their singleness is viewed differently than that of women. Marriage is considered the social norm for all Indonesian religious and ethnic groups, and motherhood plays a central role in women's lives (Situmorang 2007, p. 288; Brenner 2006, p. 2). Compared to other countries in East and Southeast Asia such as Thailand, Myanmar, Singapore, and Malaysia, which have high levels of nonmarriage, the level of nonmarriage in Indonesia remains low. In addition, in Indonesia, there is the persistence of early marriage. Teenage marriage is still prevalent in Indonesia, although the rate is decreasing gradually (Jones 2004, pp. 6–31). Commonly, young people are married off by their parents after their parents find out these young people are having sex (Utomo and McDonald 2009, p. 133). Yet, there are changes in marriage as a form of relationship and a type of behaviour in Asian countries, including Indonesia. There is a growing trend to delay marriage, especially among youth who prioritise education and employment over marriage, and some people do not marry at all (Smith-Hefner 2019, p. 135; Jones 2007, p. 454). In urban areas such as Jakarta, there is an increasing number of women who delay marriage. Numbers of never-married women are also rising, although the proportion is relatively small compared to other countries (Situmorang 2007, p. 287).

Generally, single women are viewed negatively by Indonesian society. A single woman cannot escape the question, "when will you marry?" from family, friends, acquaintances, and neighbours. This question does not stop until the woman in question finally gets married. Remaining single is not an option for women, because women who remain single are considered not to have succeeded in obtaining a husband. Although there is an increasing tendency to delay marriage in Indonesia, singleness for women is still seen a temporary state which will end in marriage and motherhood (Brenner 2006, p. 1). Women's status in Indonesia is accorded on the basis of their relationship to men and mediated by marriage to men.

The institution of marriage in Indonesia has historically been a site for the expression of multiple interests: for example to have children, to allow sexual expression, to maintain family values and traditions, and to protect morals, among others (Katjasungkana 2004, p. 154). However, as Katjasungkana (2004, p. 154) points out, none of these interests relate

to women's interests. During Suharto's rule, for instance, the institution of marriage was the extension of state gender ideology, placing women in the service of husbands and the nation. Women should be ready to perform their duties at any time. Robinson (2009) provides an example of one woman, the wife of a civil servant, who, with a heavy heart, had to abandon her sick child in order to accompany her husband to attend a certain event because her husband's superior had instructed him to do so and it was compulsory (Robinson 2009). This shows that despite motherhood also being part of women's roles in Suharto's New Order, its place is secondary to women's role as obedient wives who have to serve husbands and the nation first and foremost. Day (2006, p. 148) notes that since the late 1800s, the marriage question has been considered "one of the most important and disputed areas of debate over how to construct modern Indonesia". It was argued that a modern and independent Indonesia should be based on monogamous marriage, which triggered protests from Indonesian Muslims who practiced polygamy. Other issues related to marriage at the end of the twentieth century include arranged marriage (Day 2006, p. 148). Meanwhile, contemporary issues regarding marriage include the delaying of marriage among young people (Smith-Hefner 2019). Smith-Hefner (2019) observes that today's Indonesian Muslim youth tend to view marriage as a religious requirement; getting married is part of being a complete Muslim. Young people also consider marriage a moral obligation to their parents.

### 3. Challenging Patriarchal Marriage

Kartini (1879–1904) can be considered the first woman to challenge the condition of women and their relationship to the institution of marriage. Kartini questioned the limited options for women, arguing that women can opt for none other than to join the institution of marriage (Katjasungkana 2004). In letters sent to her friends in Europe, Kartini wrote that women were required to get married in order to be admitted as humans. Women who opted not to get married would be considered a "temptress to men" (Katjasungkana 2004, p. 155). Kartini rebelled against her situations. Kartini's father had brought her up to become Raden Ayu, the title of a high ranking official's wife. As she wrote in one letter, "I long to be free, to be able to stand alone, to study, not to be subject to anyone, and, above all, never, never, to be obliged to marry. But, we must marry, must, must" (Kartini 1985, pp. 33–34). Katjasungkana points out that more than a century later, marriage in Indonesia is still considered an obligation for women. Marriage has become "the manifestation of religious teachings in Indonesia" (Katjasungkana 2004, p. 156). While the Reformation of 1998 creates new freedom for women to express themselves, this new freedom of expression post-1998 has also given voice to a range of groups, including Islamic groups.

As a result of the Islamic resurgence in Indonesia, attitudes toward courtship and marriage have also undergone several changes. Among young people, for instance, there is a pattern of change by young educated Javanese who embrace Islamic courtship (Smith-Hefner 2005). In addition, there is a shift of attitude from "less doctrinal Javanese mores towards normative and text-based Muslim sexuality". This shift in attitude regards premarital sex as both morally unacceptable and sinful (Smith-Hefner 2006, pp. 156–157).

Despite the emergence of conservative Islamic groups who have lobbied the government to implement more restrictive laws to control sexual behaviour and public morals through legislations (Blackwood 2007), single women seek strategies to express their sexual desires. In Mataram, for instance, single women practice premarital relationships in the form of clandestine courtship and cohabitation. They keep their relationships hidden to avoid public condemnation and stigmatization, since they do not conform to the sexual ideal expected of women (Bennett 2005b).

Through her literary works, Utami takes part in challenging the construction of women in Indonesia. Her first novel, *Saman* (1998), portrays female characters that differ from the classical representations of women's feminine roles as wives and mothers. Utami's female characters are women who "transgress the normative boundaries of acceptable sexual and social behaviour" (Campbell 2007, p. 42), a theme that was rarely raised in literary works before Utami. Utami also writes essays, and in *The Single Parasite*, she wrote a manifesto

which declared her decision to never marry (2003). Utami frames her choice to remain unmarried as part of political protest toward the negative portrayal of unmarried women in Indonesia. In her words: "I want to prove that being unmarried is OK" (Utami 2003). She stresses that remaining single is an act that confronts the negative cultural views of unmarried women.

## 4. Framework

This paper considers discourses on marriage and sexuality in two texts by Utami to unravel how the concept of compulsory heterosexuality is treated in her books. In 1980, Adrienne Rich wrote "Compulsory Heterosexuality and Lesbian Experience" in the hope that feminists "examine heterosexuality as a political institution which disempowers women" and challenge it (Rich 2003, p. 11) using Rich's concept of compulsory heterosexuality to unravel the hegemonic discourses of idealised marriage in the Indonesian context. I investigate the construction of marriage for women in post-Suharto Indonesia. Rich (2003, p. 17) views heterosexuality as a "political institution" created by men, rather than as a "sexual preference" or "choice" for women. Heterosexuality is assumed to be "natural", but is, in fact, socially and politically constructed as normative, lending weight to the idea that it is compulsory. I will apply Rich's theory of the hegemony of heterosexuality to the Indonesian context and critically examine the idealization of heterosexual marriage, which is laden with gender ideology pushed by the state and later by the Islamic conservatives and its critique in the work of Utami (Parker 2008; Smith-Hefner 2019). In Indonesia's New Order, sexuality is regulated through the state's gender ideology, which emphasises women's roles as mothers in the service of the nation to support reproductive heterosexuality under Suharto's New Order—roles which can only be fulfilled by taking part in the institution of marriage (Blackwood 2007).

Parker (2008) notes that several principles underline the notion of sexuality in Indonesia: the hegemony of heterosexuality, the inseparability of sexuality with the sex/gender system, the persistence of the ideal of heterosexual marriage, the hegemonic intersections between heterosexual marriage, and the framing of reproduction and sexuality within a discourse of morality (Parker 2008, n.p.). The hegemony of heterosexuality in contemporary Indonesia is the result of a long process of the promotion of heterosexuality since colonial times. As a result, women's sexual expression is only acceptable within heterosexual marriage, which is in line with the ideology of the family (Wieringa 2000, p. 441). There are growing concerns about illicit sex among adolescents, particularly women, since women's sexuality is always attached to morality. Other studies focus on the regulation of sexuality within heterosexual marriage, and the prohibition of specific heterosexual and homosexual acts (Blackwood 2007). Blackwood shows that the regulation of sexuality has become stricter since the 1980s. She notes that at the end of Suharto's New Order regime, conservative Islamic minorities pushed the government to apply more restrictive laws to control sexual behaviour and public morals by legislating heterosexual marriage and condemning sexual practices outside of it. Indonesian Muslim conservatives consider that illicit sex is the cause of the moral decline of society, and henceforth, there should be the application of Islamic law to ban it. As Davies (2019) puts it, "politicians, religious leaders, and others in power used Islam as the justification to get rid of any factors in society that supported sexual and gender plurality and were not in accordance with heteronormativity" (Davies 2019, p. 1017). They refer to interpretations of Islam which condemn all sexual relationships outside of heterosexual marriage, which can include premarital sex and marital promiscuity, among others. Such acts are considered zina (illicit sex), a sin that can be punished legally (Bennett 2005a, p. 18). It is women, in the middle of all this, who become the centre of heated debates and contestations regarding gender and sexuality and are blamed for the degradation of morality in Indonesia (Bennett 2005a, p. 18; Brenner 2011, p. 479; Smith-Hefner 2019, p. 18).



## 5. Gender and Sexuality in Post-Suharto Indonesia

Before Suharto's New Order regime (1966–1998) collapsed in 1998, the concepts of family and marriage were central to Indonesian women's lives. This ideology of gender circulates throughout Indonesian society and serves to regulate sexual behaviour, particularly in the case of civil servants. Under regulation PP 10/1983, (male) civil servants need to ask permission from their superiors to obtain a divorce or take a second wife. Importantly, they also need to obtain their wives' permission to marry again. It has been suggested that this regulation was enacted due to the influence of First Lady Tien Suharto, who supported the protection of government officials' wives (Suryakusuma 1996).

Hatley (2008) points out that the image of women as wives and mothers symbolised the social order in Suharto's New Order. However, Hatley states, since 1998 this image has been replaced by a plurality of female forms. Artists and performers, including women writers, have become actively involved in the contests over sexual expressions, women's bodies, and national identities (Hatley 2008, n.p.).

After the Reformation, it is the Islamist conservatives that pushed Islamic law to be applied to regulate sexuality and morality. One of the many concerns relating to morality is sexual behaviour among Indonesian youth. Smith-Hefner (2019) points out that compared to previous generations, young people in Indonesia tend to be more active sexually. Yet, they are ambivalent in response to their sexuality. Matters of sexuality are things that youth have to grapple with on their own. Most parents do not discuss sexuality openly: they only provide vague advice related to sexuality to their children (such as telling their daughters to be careful and safeguard themselves when they reach adolescence) or hope that their children find out about sexuality on their own (Smith-Hefner 2019, p. 126). Smith-Hefner (2019) outlines how—to provide a moral foundation to their children—middle class parents enrol their children in religious education both at school and through *pengajian* (informal religious classes) as early as four years of age to lay the groundwork for a strong moral foundation. As a result, Indonesian youth hold normative Islamic views on sexuality gained from years of attending religious education since childhood, such as sexual activity ideally being confined in the legal state of marriage and sexual engagement outside of marriage being considered the sin of "zina" (illicit sex) which can be a threat to one's morality. However, Indonesian youth do not always live up to this normative ideal (Smith-Hefner 2019, p. 129). The dominant discourse of sexual morality in Indonesia has a great influence not only on Muslim youths but also on Indonesian Christian youths. Christian youths should choose either to practise heterosexual abstinence or to have "double morality" to conform to the expected ideal of sexual morality. The other alternative that Christian youth resort to when faced with the dillema of choosing whether to be in the "moral' or "immoral" camp related to their sexuality is by discarding their religion completely (Mulya 2019).

Bennett (2005a) points out that although illicit sex is considered inappropiate behaviour for both men and women, it is women who are primarily forced to stick to this moral code. Women have to face double standards related to illicit premarital sex, while it is often acceptable for men to be involved in premarital sex. Engaging in sexual activities prior to marriage can destroy women's reputations. They can be shamed, labeled, excluded from the community or family, and in some areas, even abused. Therefore, women engaging in premarital sex often do so discreetly. In public, they continue to uphold "the ideal of female purity" (Bennett 2005a, p. 19).

Brenner (2011) notes that in the past two decades, the Islamic movement and the democratisation movement competed against each other to gain "symbolic control over public morality", so that gender and sexuality have become key pillars of political power. Heated public debates related to women also involve women from diverse groups as participants (Rinaldo 2011). In public debates about pornography and polygamy, women's groups based in political Islam and Islamic traditions have differing interpretive approaches to Islam's constructions of morality which have a bearing on the future of Indonesia (Rinaldo 2011, p. 557).

One example of highly contested and heated public debates is the controversy surrounding the proposed Anti-Pornography Bill of 2006, which became known as a "culture war" (Bayuni 2006 cited in Allen 2007b, p. 101). The Bill was originally drafted in the 1990s, but was widely discussed in 2005–2006, during Susilo Bambang Yudhoyono's Presidency (Pausacker 2008, *Inside Indonesia*: n.p.). Controversy about the Bill was triggered by the broad definition of what constitutes pornography and pornoaction. For instance, according to one article, women can be imprisoned if their clothing is considered to reveal "sensual parts of the body" (Allen 2007b, pp. 101–2). Weinstraub (2008) argues that women's bodies have become the focal point for public debates. Some critics argue that the performances of Inul Daratista, a singer/dancer, constitute a form of pornography. Weinstraub (2008, p. 367) argues that Inul's goyang ngebor (the drilling dance) was considered "pornographic" in 2003, pointing out that Inul's dancing body became a central symbol for debates about culture, religion, and politics after Suharto's downfall. At the time, Amidhan, the leader of Majelis Ulama Indonesia (MUI—the Indonesian Council of Ulamas), cited a report that Inul's goyang ngebor had provoked a man to rape. Amidhan blames women for male sexual violence.

Other critics have pointed out that the Bill was used to distract attention from the government's failure to eradicate corruption, to improve the education and health sectors, and to address poverty and unemployment (Allen 2007b, p. 103). The Bill was promoted as an attempt to "protect" women. Nevertheless, women activists view the Bill as having the potential to increase violence against women if it becomes law. One notable protest was a huge street rally by women's groups on International Women's Day in 2006, which included Shinta Nuriyah Wahid (wife of former president Abdurrahman Wahid), Nia Dinata (movie director), Ratna Sarumpaet (playwright), and Ayu Utami (writer) (Allen 2009, n.p.).

Allen (2007b, p. 112) argues that resistance to the Bill indicates the fearfulness of many Indonesians, including Muslim Indonesians, about the tendency toward an Islamised state, as this has the potential to threaten cultural diversity. For example, ordinary Balinese men and women who bathe in the river will be considered to be engaging in obscene acts, which, therefore, constitutes "pornography". On an online blog posted by a Javanese Muslim, the practice of outdoor bathing was perceived as "disgusting and pornographic" (Bellows 2011, pp. 218–19). Allen points out that the term "pornography" in Indonesia has a straightforward meaning; anything to do with sex is considered pornographic, including a text (Allen 2007a, p. 29). The implications of the Anti-Pornography law, which finally passed in 30 October 2008, produce the state as a "peeping Tom" who monitor sexual affairs, illicit sex, and cultural performances suspected to be "pornographic" (Bellows 2011).

According to Bellows (2011), there has been a transformation in the figure of the peeping Tom in contemporary Indonesia. Before cell phones and handycams, the peeping Tom spied on places usually used for illicit sex such as the beach or old building sites, but then captured and shamed the couples by spreading the news through the neighbourhood. In the digital era, the peeping Tom uses a camera to capture couples having sex or steals personal videos to post online. The couples' affairs are made public, while the peeping Tom remains invisible (Bellows 2011, pp. 224–25). Allen (2009, n.p.) asserts that public debates, whether in support of or opposition to the Bill, show new freedom after the Suharto era, allowing various voices, including women's voices, to be heard in public life.

Another issue that has become a matter of public debate is polygamy. Brenner (2006) argues that democratic changes brought by the Reformation have encouraged polygamists to publicly promote polygamy. Puspo Wardoyo, President of the Indonesian Polygamy Society (Masyarakat Poligami Indonesia), has triggered public debate with his Polygamy Award, a ceremony for "successful polygamists". The atmosphere of democracy has become a reason for polygamysts to be "transparent" about their practices. However, antipolygamy activists contend that the practice of polygamy is both undemocratic and oppresive to women (Brenner 2006, p. 166). Van Wichelen (2013) has also explored the public debates on polygamy in post-Suharto Indonesia. She suggests that the discourse on

polygamy and its contestation by women's groups cannot be separated from the influences of nationality, religion, modernity, and globalisation.

Muslim feminists in Indonesia not only contribute to public debates about women's rights but also take initiatives to exercise a degree of control in women's lives (Van Doorn-Harder 2008). These debates and initiatives relate to women's status in marriage law, polygamy, and reproductive rights, among others. Van Doorn-Harder (2008, p. 1039) points out that Indonesian Muslim feminists carefully use arguments based on the reinterpretation of the Qur'an to protect women's basic rights. Rinaldo (2011) explores the involvement of women activists as participants in the debates about gender and sexual morality. She focuses on Muslim women activists in two Indonesian Muslim organisations, Fatayat Nahdatul Ulama (the Awakening Ulama/NU) and the Prosperous and Justice Party (PKS). The former uses a contextual and revisionist approach to Qur'anic texts, while the latter uses a literalist interpretation of the Qur'an. In regard to issues of polygamy and the Anti-Pornography Bill, women activists from these two organisations have different points of view. While PKS women have a range of views on polygamy, they are reluctant to ban it. Fatayat women view polygamy as a form of discrimination against women and argue this contributes to wider violence against women (Rinaldo 2011, pp. 552–55). Siti Musdah Mulia, a leader of Fatayat NU, has argued for reform of the conservative Islamic Law Compilation (Budiman 2011, p. 85). Budiman (2008, p. 89) notes that the struggle of Muslim feminists is supported by male Muslim intellectuals and ulamas who work together with women activists to disseminate alternative interpretations of Islamic texts (the Qur'an and Hadith) to counter dominant patriarchal readings of the texts.

In an era of decentralisation, there are restrictions on women through various sharia laws at the local level (Brenner 2005, p. 116 cited in Day 2006, p. 149). Robinson (2012) points out that gender relations become key elements of these local public debates. There are regulations over women's clothing in some regions in Indonesia, which have triggered protests from women activists. There is also debate about whether Islam provides for women's leadership by Islamic organisations, while at the same time the new-found freedom of expression creates opportunities for women to become candidates for the House of Representative locally and nationally (Robinson 2012, n.p.). Pausacker states that the enforcement of Islamic values has even penetrated legislation and judicial decisions (Pausacker 2012, p. 1). This can be clearly seen in cases such as the imprisonment of the editor of the Indonesian Playboy magazine, Erwin Arnada, for "violating the norms of propriety" in the community, despite the fact that the content has been "toned down" for the Indonesian market (Pausacker 2012, p. 1).

## 6. Women and Literary Works

Post-New Order freedom has influenced literary production, particularly in the work of women writers. Women writers not only struggle for literary emancipation, but also for freedom in public/private life (Day 2006, p. 150). Bodden and Hellwig (2007, p. 1) point out that after Suharto's resignation, writers and filmmakers flourished as a result of the new-found freedom of speech. Utami is one of those women writers whose work became part of heated debate, after her first novel *Saman*, which expresses ideas about female sexuality, was published in 1998 (Bodden and Hellwig 2007, p. 2). Not only have her literary works been the subject of academic study and literary criticism, but *Saman* was also publicly considered controversial, as female writers have never written in such an outspoken way in portraying sex and sexuality. Bodden and Hellwig (2007, p. 2) suggest that Utami's novel represents an "unprecedented space for writers, artists and activists of various kinds to explore new horizons" (Bodden and Hellwig 2007, p. 2).

*Saman* was published about a month before Suharto's downfall. The success of the novel inspired other women writers to write about contemporary issues of identity and sexuality (Campbell 2007, pp. 42–43). Following the success of *Saman*, several new women writers began to write openly about female sexuality in their novels. Writers such as Dewi Lestari, Djenar Maesa Ayu, Dinar Rahayu, and Nova Riyanti Yusuf, among others, have

written about controversial issues such as homophobia, lesbian relationships, transsexuality, and sadomachism (Bodden and Hellwig 2007, p. 6). The literary works of these women writers are often referred to as sastrawangi (fragrant literature). The term of sastrawangi is a cynical term, used by male literary critics to undermine the works of women writers. As Bodden and Hellwig (2007) note, the term "reflects the fragrance of sastrawati, the young and attractive bodies of its female producers" (Bodden and Hellwig 2007, p. 7). Thus, women are relegated to the body in much male criticism. Taking aside the quality of their works, the focus of attention is women writers' physical attribution which is associated with "fragrant" and "sexy" (Arnez and Dewojati 2010, p. 11). As a result, the literary works of women writers are undermined. Hellwig (2007, p. 128) sees the term from a feminist standpoint, arguing it reveals the effort of the patriarchal hegemony to reduce women writers to their physical appearance, rather than assessing the merits of their work. Such writing is considered not to provide "the correct moral and intellectual teachings", nor to contribute to the "building of the nation" (Allen 2007a, pp. 26–30). The assumption that women writers such as Utami only write about sex overlooks the social criticism evident in their works. In *Saman*, for instance, Utami addresses the struggle of marginal farmers against big capitalist companies. Themes on domestic violence appear in Djenar Maesa Ayu's writing, which frequently addresses dysfunctional families (Bodden 2007). Women writers are criticised for being preoccupied with sexual activity in their novels. However, Allen contends that the work of women writers subverts dominant patriarchal notions of morality and womanhood in Indonesia (Allen 2007a). The political dimensions of women's writing are overlooked in the critics' preoccupation with women daring to write openly about sexual relationships and sexual identities.

Hellwig (2011) analyses literary works by women since 1998 which depict marriage and sexuality and portray shifts in the concepts of marriage, sexuality, and gender roles. In her findings, Hellwig describes how female fictional characters function as agents of change. She examines codes of conduct and cultural practices in literary texts by four women writers: Ayu Utami, Alia Swastika, Fira Basuki, and Stefani Hid. Her analysis and close readings of literary texts show that there have been changes in the discourses on marriage, family, sex, and sexuality in post-Suharto Indonesia. Democratisation that arose in the post-Suharto era has provided space for self-actualisation and personal choice; yet, marriage is still considered "sacrosanct", and representations of alternative ways of life such as self-chosen singlehood, for instance, still trigger conflicts (Hellwig 2011, p. 17).

Day argues that in post-Suharto Indonesia, the most important issue is the meaning of the term "freedom" (Day 2006, p. 147). As Day puts it, "Indonesians are crazy of [sic] many kinds of freedom (kebebasan)–freedoms that are subjective and sexual as well as public and political" (Day 2006, p. 148). The democratisation of Indonesia and the new freedom of expression that it is assumed to have been gained by Indonesians after Suharto's resignation has also been accompanied by a resurgence of Islamist conservatism. Radical Islamic groups have arisen to enforce "morality". There is also a tendency for radical Islamic groups to use violence in their actions. The Islamic Defender Front (FPI), for instance, often uses violence when conducing raids on brothels and illegal gambling. Not only does it use violence on the street, but FPI also resorts to violent acts anywhere. FPI spokesman Munarman threw water at a sociologist during a live discussion on television (The Jakarta Post 2013a). However, the government does not take stern action against Islamic organisations who promote violence. Indonesian Home Affairs Minister Gamawan Fauzi even stated that FPI is "an asset to the nation", and that the government plans to cooperate with such organisations in development programs (The Jakarta Post 2013b).

The diversity and pluralism of Indonesian culture is also threatened by the politicisation of differences in religion, race, and ethnicity. For example, during the gubernatorial election of Jakarta in 2012, singer-turned-politician, Rhoma Irama, a supporter of the incumbent governor, campaigned in the mosque against voting for a non-Muslim leader. At the time, gubernatorial candidate Joko Widodo, whose vice-gubernatorial candidate was Basuki Cahaya Purnama, a Chinese Christian, countered the attack by saying he was a

"big fan" of Rhoma Irama and that one of his favourite songs is 135 Juta (135 millions), which celebrates diversity in Indonesia (The Jakarta Globe 2012). Joko Widodo won the election despite the alleged attack using issues of religion, race, and ethnicity. However, the issue of religion arose again with the appointment of the subdistrict head of Lenteng Agung, Jakarta, Susan Jasmine Zulkifli. Muslims protested that Zulkifli should be moved to another region, because she is a Christian in a Muslim community. While Governor Joko Widodo refused to remove Zulkifli on religious grounds, Home Minister Gamawan Fauzi argued she should be removed because the protests against her leadership would affect her performance in the role (Nurbaiti 2013, the Jakarta Post). The use of social media platforms for political campaigns during elections further exacerbates the polarisation. Lim (2017) asserts that social media not only facilitates freedom of expression, but also allows "freedom to hate"; individuals are able to voice their opinions and to hush the voice of those who have different opinions or different political preferences from their own (Lim 2017).

## 7. Method

Many studies have focused on marriage in Indonesia, in areas as diverse as marriage and religion (Seo 2013); polygamy (Brenner 2006); "informal" marriage (Brickell and Platt 2013); transsexual marriage (Wieringa 2013); marriage law and human right issues (Bedner and Huis 2010); ethnic diversity and marriage behaviour (Buttenheim and Nobles 2009); transitions to marriage among middle class youth (Nilan 2008); and marriage among educated youth (Smith-Hefner 2005). However, there are few studies that analyse marriage and single women. One of the few studies was conducted by Situmorang (2007), who analysed the trends and patterns of never-married women. She focuses on the lives of unmarried women in their thirties and forties in Yogyakarta and Medan. Utami's literary works offer opportunities to analyse the links between marriage as a form of compulsory heterosexuality in Indonesia and the stigma faced by unmarried women who resist.

In this paper I apply discourse analysis to Utami's texts. According to Paltridge (2012), discourse analysis examines "patterns of language across texts and considers the relationship between language and the social and cultural context in which it is used" (Paltridge 2012, p. 2). I use Adrienne Rich's theory of compulsory heterosexuality as a tool to analyse the text and relate Rich's theory to the construction of marriage in Indonesia. I also apply intersectional analysis drawing on Chris Klassen's work on the importance of religion in the analysis of women's position in society. I translate the texts from Utami's two books, *The Single Parasite* and *Confessions*.

## 8. Findings

In *The Single Parasite* Utami declared that she would never marry. After writing two novels, *Saman* (1998) and *Larung* (2001), Ayu Utami published her third book in 2003, which was republished in 2013. *The Single Parasite* comprises 33 short essays which were taken from her earlier writings in JAKARTA JAKARTA Magazine and djakarta! Magazine in 1999–2003. The essays are divided into three parts: Kehidupan (Life), Seks, Jender dan Kapitalisme (Sex, Gender and Capitalism), and Politik dan Negara (Politics and the State). Utami also includes an epilogue to the book. *The Single Parasite* received wide critical and commercial acclaim. One review says that reading the essays in *The Single Parasite* provides a "unique and terrifying experience", since Utami addresses topics related to women and marriage that are deemed a "hypersensitive issue" in contemporary Indonesia (DVRG Magazine 2013). The book was republished in 2013 alongside her later book, *Confessions of a Former Single Parasite*, in which she outlines the reasons behind her change in thinking about marriage.

The title of the book is taken from a term used in Japan to describe single women who are considered to enjoy an easy life. They generally live with their parents while having good careers and a comfortable salary at the same time (Utami 2003, p. 115; Jones 2004, p. 23). Utami uses the term to refer to her status as a single woman who lives with her

parents, and considers that for her mother, Utami might be considered a "parasite" because she does not have to think about the cost of living (Utami 2003, p. 115). The book reveals Utami's concerns about a range of social issues in Indonesia from religion, sexuality, and marriage to politics and the state. She discusses everyday incidents and provide views which challenge Indonesian patriarchal culture in short essay form. However, her writings are far from being too serious. There are elements of humour here and there. For example, she writes about how Indonesian people are obsessed with religion:

> "... finding out about someone's religion is one of the top five things Indonesian people would like to know (the other four are age, marital status, ethnicity, and employment–complete with the salary if needed)" (Utami 2003, p. 10).

Utami writes of her experience as a single woman, routinely questioned and scrutinised if they are not yet married. In the epilogue, she bluntly states that she will never marry, providing 11 reasons for this decision (Utami 2003, pp. 168–76). The political stance is a challenge to the social construction of women's necessity to marry in the Indonesian context.

Rich's concept of compulsory heterosexuality which centres on how female sexuality is suppressed and regulated in heterosexual relationships with men. Heterosexuality becomes a political institution (Rich 2003, pp. 11–12). Utami's essays in *The Single Parasite* give a detailed elaboration of the discourse of compulsory heterosexuality in Indonesia as it is embodied in the institution of marriage. In the essay "Sex Education in School", Utami recalls her experience as a student in sex education classes. Male and female students were separated to different classes. What she took from all those sex education classes was that sex had only to do with reproduction. Reproduction organs were all about the function to reproduce, while explanations about the pleasure of sex were ignored. Indeed, teachers showed students frightening images of sexually transmitted diseases (STDs) which, in Utami's opinion, were meant as a form of shock therapy to scare students off of engaging in sexual activity. She wished she had had sex education classes that had been delivered in a way that would make students able to appreciate sex just as they appreciate music or film (Utami 2003, pp. 63–65). However, those classes she had attended during her school years reflect the idea that sex is only for heterosexual adults in the bounds of marriage. Openly discussing sex and sexuality in Indonesia is considered taboo. In "Let's Have Children in Town", Utami discusses how the discourse of sex and sexuality is often wrapped in metaphoric language even when consulting doctors. To ask whether a female patient is sexually active, a doctor uses the question, "do you have children?". Utami argues that this "moralism" has penetrated bureaucracy, and states that she often uses "taboo" words related to sex and sexuality in order to not be conquered by euphemisms, openly using words such as penis, vagina, and orgasm. Utami states that gender bias can be clearly seen whenever Indonesian women have to deal with doctors. The question "Mrs or Miss?" means the doctor is asking whether the patient is sexually active or not. If the answer is "Miss", the doctor will assume that the patient is not sexually active, because only those who are married are presumed to engage in any kind of sexual activity (Utami 2003, p. 92). Single women's sexual conduct is even scrutinised through "medical check-ups" when they want to enter a certain profession. If they aspire to be policewomen, for example, female applicants must go through "a virginity test" (Davies 2018). Merely stating that they "do not have children" is not enough; applicants should undertake vaginal and hymen examinations to determine that they are still virgin. Although it is claimed that the virginity testing is not conducted anymore, vaginal and hymen inspections still continue in the guise of "reproductive health checks" (Davies 2018).

In the essay "Polygamy, Government Regulation No. 10 and Hypocrisy", Utami bluntly states "I am anti-polygamy" (Utami 2013, p. 171). She observes that polygamy is reasonable in a very patriarchal society that assumed women cannot live without the protection of men. Utami initially wrote on the theme of polygamy in response to the planned removal of a 1983 government regulation which banned polygamy for Indonesian civil servants, stating that she did not agree with its removal, calls for which became

stronger after the downfall of Suharto's New Order. In the post-Suharto era, the supporters of polygamy attempted to frame polygamy as a personal and private matter (Utami 2003, pp. 164–66). Although she disliked Suharto, Utami preferred Suharto's stand on monogamy to then-Vice President Hamzah Haz's sanction of it. Utami argued that polygamy is laden with patriarchal values which considers men as needing to "help" and "protect" women through the practice of polygamy (Utami 2003, p. 171). She felt disappointed when the Minister for Women Empowerment in the administration of President Abdurrahman Wahid did nothing when Islamic groups proposed removing the prohibition on officials having more than one wife (Utami 2003, p. 171). After the fall of Suharto, government officials who had taken multiple wives became more visible. Their polygamous marriages became known to the public when they were revealed to be involved in corruption. The Jakarta Post (2013c) reports that polygamy and graft became a "most convenient marriage", as corrupt government officials took second or third wives to hide their crimes, buying cars, houses, and land in the name of multiple wives.

Utami states that there is no place in Indonesian patriarchy for single women. Even an identity card only provides three choices: married, widow/widower, and not yet married. The status of unmarried people is not taken into consideration. The question "are you married?" cannot simply be answered by "no" because it will trigger another question: "why not?". The questioner always needs a "rational" explanation for such an answer and always provokes the response, "not yet" (Utami 2003, p. 93). This means that singlehood is not a choice, but only regarded as a temporary stage. The possibility of getting married remains open; single people are presumed to end their singleness at some future time. In an essay entitled "In a Train Carriage", Utami tells of her experience as a single woman travelling by train to another town. Even in a train carriage, she cannot escape questions about her single status from a stranger. Utami points out that a single woman who is not married is deemed to hold a "minus" value, because marriage is equated to the neutral position of "zero" in Indonesia (Utami 2003, pp. 90–91).

In the epilogue to the book, Utami declares that she will not marry, framing this as a political act. As one of her reasons, she states that she did not want to contribute to the growth of population in Indonesia: "I do not want to reproduce since (Indonesia) is already overpopulated" (Utami 2003, p. 175). While Utami never intended to provide reasons for purposely remaining single, she was constantly forced to formulate answers to the question that never went away: "Something that I consider a normal thing now transforms into a political stance" (Utami 2003, p. 176). However, as *Confessions* shows, Utami's political stance about marriage shifted in the years after she published *The Single Parasite*.

Compulsory heterosexuality could render women powerless, thus, Rich encouraged heterosexual feminists to examine and challenge heterosexuality as a political institution (Rich 2003, p. 11). Utami addresses how single women who do not follow the normative ideal of women as wives and mothers is observed with eyebrow raised. The state of being outside of the institution of marriage is continuously questioned. The conversation on a train between Utami with the other passenger confirms it. Feeling astounded to find out about Utami's age (in her thirties at the time), a young man exclaims, "Gosh! If you were in the village, you would have had three children!" (Utami 2003, p. 90). The conversation represents how a single woman is supposed to have already married and procreated by a certain age. Compulsory heterosexuality also presents itself in terms of sexual activity, which is only legitimate for married couples. Unmarried couples and teenagers are not allowed to engage in sexual activity. This is obvious when Utami discusses her sex education classes during high school. Sex education classes were conveyed in such a way to only show the functions of sexual organs, and sex was reduced to procreation. Sexual activity is portrayed as something dangerous for teenagers daring to explore this restricted area. Horrible pictures of sexual transmitted diseases (STDs) were shown to discourage students from engaging in activities which are only "legitimate" for adults engaged in a marriage (Utami 2003, pp. 63–64). This also reflects Parker's argument that sexuality in Indonesia is framed within "morality" and that it is considered "immoral" to

engage in sexual activity before marriage (Parker 2008). Therefore, safe-sex campaigns to prevent STDs by using condoms have faced barriers in the name of "morals and religion". Conservative groups who banned the campaigns are unconcerned about the spread of STDs because they believe "morals and religion" will be effective to prevent it. Instead, they are more concerned that safe-sex campaigns will encourage "free sex" (sexual activities out of wedlock). However, as Utami (2003, pp. 50–51) argues, "morals and religion" are not enough to ward off STDs. "Morals and religions" do not stop Indonesian men from visiting prostitutes or having sexual infidelity outside of their marriage. In this case, women's position is weak because they can be infected by their own husbands who might be committing adultery outside the bounds of marriage. Hence, Utami asserts, safe-sex campaigns that promote condom usage need to be supported in order to protect women. Her manifesto about not getting married acts to challenge the view that marriage is compulsory for a woman, and so challenges compulsory Indonesian heterosexuality.

In 2011, Utami married her long-term partner, photographer Eric Prasetya, at a church in Bogor, West Java, Indonesia. Their marriage was only registered in the church. In Indonesia, marriage is usually registered by the state after religious marriage since civil registration is required "for the validity of marriage regardless of religious background" (Seo 2013, p. 79). The news of her marriage spread through social media, such as Twitter and Facebook. Through her Twitter account, @BilanganFu, Utami tweets, "my choice of non-marriage is a political act, my choice of marriage is an act of faith" (Octavia 2011). Utami's *Confessions of a Former Single Parasite* (2013) has a parallel theme to *The Single Parasite: Sex, Sketches and Stories* and explains her shift in position in regard to her proclamation that she would forever remain single. Confessions is described as "an autobiography on sexuality and spirituality written in the form of a novel" (The Jakarta Post 2013d). Utami uses the title Pengakuan (Confessions) to reference how she was inspired by the autobiographical work of a Christian thinker, Saint Augustine, and his work Confessions (A.D. 397). The book is divided into three parts: a Woman Who Lost Her Virginity and Became an Adulterer, A Girl Who Lost her Faith, and On Her Way Home. *Confessions* is about a woman called A who experiences existential angst, feeling at odds with the morality upheld by her society. A considers such morality problematic, especially for women. She rebels against social and religious values which discriminate against women and is determined to define her own values. She explores her sexuality, distances herself from religion, and, in the end, finds her own form of spirituality. A despises the concept of virginity in a society which requires only women to preserve their virginity for their future husbands, noting men are never asked to remain virgins for their future wives. She watches a television program about a famous male singer who divorced his wife not long after they were married because he suspected she was not a virgin. "My mother once told me that a woman is like porcelain," she writes—once it is broken, it no longer holds any value. In A's view, it is ridiculous that virginity reduces a woman to a mere membrane (Utami 2013, p. 35) and erases the importance of the virginity from her life before she reaches the age of 20. Women's virginity is deemed a greater value compared to that of men in Indonesia. There are various terms which degrade women who are no longer virgins outside of marriage, such as rusak (broken or damaged), hancur (crush or pulverized), murah (cheap), and gampang/mudah (easy), among others. Once women suspected of having premarital sex, their reputation is damaged (Bennett 2005a, p. 19).

A is also highly critical of religion. Born in a Catholic family, she finds that there is something wrong with the way Catholicism treats women. She cannot comprehend how a religion which was supposed to treat men and women equally placed men in a higher position than women and could not accept that only men can be leaders and that women are not allowed to be ordained as priests in the Church. In addition, she observes that religion has become a source of violence, with a history of violence committed in the name of religion. These lead her to proclaim: "I will not subscribe to religion anymore" (Utami 2013, p. 36). Since A no longer believes in religion, she does not feel guilty when she starts having sex with her boyfriend, Nik, who comes from a different religious background.

"Since I was outside of religion, why should I concern myself about the concept of illicit sex?" (Utami 2013, p. 36). Abandoning religion altogether is the alternative option that Indonesian Christian young people often choose when faced with the moral/immoral positions regarding their sexuality (Mulya 2019, p. 60). It is part of their resistance toward the dominant discourse on sexual morality. The character A cannot live with two value systems that collide with each other. She cannot be a good Catholic and practice illicit sex, which is deemed an immoral act, at the same time. She chooses the latter because she thinks it is unethical to do both (Utami 2013, p. 45). Mulya (2019) points out that being outside of a religion enables the youth to redefine morality, which results in "a more personal and negotiated ethical approach to sexuality" (Mulya 2019, p. 60).

The second part of the novel describes how A's childhood shapes the way she views the world. She describes how her two aunts became bitter toward other women because they failed to live up to society's expectations that they would get married. She starts to see that unmarried women are jealous of married women, especially at school, where female teachers who were "old maids" (Utami 2013, p. 114) treated beautiful female students badly when they made mistakes. A sees it herself and feels fortunate that she is not considered beautiful enough to be mistreated by her unmarried female teachers (Utami 2013, p. 115). She sees nothing wrong with being an "old maid". However, women are only considered respectable when they are married, so that the institution of marriage, and women's relational status to men, uphold social norms. Outside marriage, women have no "status". A starts to think that not getting married could be a political act that rejects the social norms that eliminates the dignity of women.

In the third part of the novel A is determined to stick to her decision to be outside of the institution of marriage. She makes a manifesto, proclaiming she would never marry. This attracts huge attention. She receives many questions and suspicions that she might be a lesbian or have traumatic experiences with men. Others thought she might come from a broken family. A does not care about it. However, she thinks about how her choice might affect her mother. Her mother would not comprehend her daughter's political stance, namely, that she wants to liberate women from the social pressure to marry and to release them from dependency on men. Moreover, her father also opposed her decision at first, but gradually he came to terms with it. At her sister's wedding, her father even announces to the guests that this would be the last wedding in the family because his youngest daughter has chosen not to get married. Other relatives view A's decision as a mere temporary matter; they believe A will eventually get married. However, her father firmly believes that she will never marry (Utami 2013, pp. 225–26). A eventually meets Rik. They have much in common; for instance, Rik has also chosen not to marry. He just wants to be free. His decision is not based on a political stance like A, but still, she finds it strange and refreshing that she could find such a man in Indonesia (Utami 2013, pp. 240–41). A realises she cannot continue to cheat on her partners like she used to in her previous relationships, because it is unethical. She wants a relationship based on respect.

Against a backdrop of an increase in violence in the name of religion, minority groups face difficult times, and A feels she has to do something to challenge violence in the name of religion. She not only wants to stand for those who are oppressed, but also to show solidarity with the Church, yet there are philosophical barriers. Having criticised Catholicism, A feels that she is no longer a part of that community. Her involvement with the Church community leads her to think about marriage. However, she cannot accept the fact that Catholicism discriminates against women (Utami 2013, pp. 272–73). She later discovers that Catholicism actually has an egalitarian concept of marriage. A man does not necessarily become the leader of a woman or a husband the leader of his wife. It agrees with her political stance. Having found no conflicting ideas which oppose her stand, she determined to marry her partner in the Church. However, she is still against patriarchal marriage exercised by the state. Therefore, she refuses to register the marriage with the state. Indonesian marriage law, in her view, still positions men as the sole head of the family. The sacrament of marriage in the Catholic Church is an act of solidarity with her

Catholic community (Utami 2013, p. 292), a religious minority community which she has been involved with over time. She wants to continue working with them. A no longer thinks that a woman who is married or not married is a problem. "Time has changed. Women are now more daring and independent" (Utami 2013, p. 292).

In 2011, Utami had hinted at her intention to get married during an interview in which she stated that she did not mind getting married under Catholicism, because there was no obligation for men to be heads of the family. "I probably will do it in the near future", Utami said (Fimela 2011). The fact that she did marry contradicts her previous political proclamation that she would not join the institution of marriage. Utami argued that she was now more concerned about the rise of intolerance sweeping the nation (The Jakarta Globe 2012, n.p.).

Utami's change in attitude surprised many critics, and Utami, therefore, felt the need to justify her decision. She notes that her decision to get married had made a single mother feel depressed over the news (The Jakarta Post 2013d). There were a lot of comments about Utami's marriage on social media. "Everybody will question her previous declaration to never marry", comments Octavia, a Muslim blogger, to which another blogger defends Utami's decision. "It is part of her spiritual journey. In my humble opinion, there are things that can be changing in this kind of spiritual search" (Octavia 2011). Another fan expresses her disappointment in Utami's attitudinal change. "I stopped being a fan", states Anies Syahrir, an activist and former journalist, in her blog, as she considers Utami someone who is not able to stay true to her principles. In her opinion, Utami's attempts to explain her shift in position is worthless. "She has destroyed what she has built for years" (Sjahrir 2013). However, it could be argued that Utami did not betray her previous critique of marriage in *Confessions of a Former Single Parasite*. She remains determined to destroy some of the sources of problems for Indonesian women: the concepts of virginity and marriage. She wants women to be free from pain and jealousy just because they do not get married. A despises the concept of virginity, viewing it as a "commodity" that can only be traded in marriage, which degrades women. She fights against the social construction of virginity as something that has a value to be exchanged, arguing for the diminishing of the exchange value of virginity. "The concept of virginity is just made up by those who are fond of exchanging the value of goods and making profit from it", she declares (Utami 2013, p. 163).

Themes of religion and spirituality are obvious in *Confessions*. The main character, A, is searching for the meaning of religion. She wonders why religion has perpetuated patriarchal arrangements, rather than viewing men and women as equal human beings who should be treated without discrimination. This injustice toward women is also revealed in the discourse of morality, which A observes as targeting women. "I cannot accept values which are unfair" (Utami 2013, p. 35). Since religion also treats men and women differently, A decides to stop subscribing to religion, which is, in her case, Catholicism. She cannot accept that her religion could be the source of hostility toward women. Her decision to leave Catholicism was a result of her observation that Catholicism was unjust toward women, for example, in that they cannot be Catholic priests. In her view, her religion made men superior to women. Hence, A distanced herself from Catholicism, but she still has interest in religion and spirituality, considering Protestantism superior in relation to gender, since women can be pastors. However, she had a special attachment to Catholicism, especially to Mary, mother of Jesus, saying that "Mary is my spiritual mother... She had empathy to those who suffer, and she persuaded her son to help them" (Utami 2013, pp. 156–57). Despite her crisis of faith, she could not stop her interest in religion, and she still read the Bible and got involved with the Church community. At the time, (Muslim) religious paramilitary groups were involved in regular disturbances toward minority religious groups in Indonesia; although A pays little attention to religion, she still feels part of an oppressed religious minority (Utami 2013, p. 272). This leads to her thinking about marriage in the Catholic Church. A wants to participate in her community, yet, feels she is an outsider because she is living with a man without being married. While nobody

bothers her about her life choices, A feels uncomfortable about it. By getting married, A can truly feel that she is part of her community. She does not think that she has changed. Rather, marriage is not as important an issue to her anymore. It is part of her negotiation of her spiritual and political development; she sticks to her convictions by not registering her marriage with the state, because marriage still positions women as lower than men in the family.

Saras Dewi, a philosophy lecturer at Indonesian University ([Dewi 2013](#), p. 1), states that Confessions advocates for a more equitable world for women, with its serious message framed by elements of humour. For Dewi, the book makes readers imagine an alternative universe where women are truly independent, society does not mask its face behind the facades of morality, and justice is not merely a concept argued by philosophers in books ([Dewi 2013](#), p. 1). Meanwhile, one writer argues the book does not provide enough explanation about the character's attitudinal deep shift from a political stance against marriage to an eventual entry into the institution of marriage based on evidence that an egalitarian concept of marriage existed somewhere in Catholicism ([The Jakarta Post 2013e](#)).

Utami's second novel reveals that patriarchal culture is spread by misogynist interpretations of religious teachings, discussing the significance of the role of religion in women's lived experience. Chris Klassen states that there is a tendency for an analysis of religion to be left aside in (Western) feminist research. As a result, this excludes the lived experience of many women who practice religion and look to religion as a source of strength ([Klassen 2003](#)). In *Confessions*, A refuses religion because she considers it hostile to women, because it places women under men's control. She challenges this view by keeping a distance from religion, but then reconciles with her Catholicism when she is sure that there is no contradiction in the position of men and women. *Confessions* is the story of a woman who, in changing social conditions, and as part of her spiritual journey, is thinking about the gendered and religious dimensions of her identity together. This insight helps explain why Utami has changed her position on marriage. Therefore, Rich's theory of compulsory heterosexuality is not enough to explain the intersectionality between marriage and religion required by Klassen's argument. Utami negotiates her position as a Catholic woman who is concerned with the tendency of growing violence and oppression toward minority groups in Indonesia. Marriage is a strategic means for Utami to show solidarity toward her Catholic minority community against oppression in the name of the majority religion in Indonesia.

**9. Conclusions**

The downfall of Suharto in 1998 provided space for freedom of expression, including for women. Indonesian women had more space to express their opinions in a newly democratic atmosphere. Women contributed to and actively participated in public debates, including through creativity in the media and the arts, as women writers and filmmakers redefined gender roles and represented women in different ways. Ayu Utami is one of Indonesia's celebrated women writers who portrays women outside the stereotypes of wives and mothers. Her declaration to remain unmarried was outlined in the manifesto of *The Single Parasite*, operating as a challenge to the state's gender ideology and its framing of women's sexuality as only acceptable in heterosexual marriage and as wives and mother serving husbands and the nation, using the construction of Rich's framework of compulsory heterosexuality in an Indonesian context. Her proclamation to never marry disrupts the perpetuation of patriarchal culture, which suggests that women are only happy if they are married; she shows that being single is a valid way of life, and that Indonesian women can choose whether or not they want to marry.

Utami's shift of position in Confessions of a Former Single Parasite proclaims her belief in women having options, outlining her view that freedom to marry must also be an option. She chooses to get married in the Church after finding out about concepts which support her political stance regarding gender equality. Moreover, she feels it is high time to be more involved in her religious community in the face of a rise in Islamic conservatism, which

poses a threat to religious minority groups such as hers. The autobiographical *Confessions* shows the journey of A (the main character), which begins with the realisation that religion is problematic since it operates as the source for the marginalisation of women. However, she later reconciles with her Catholicism because she acknowledges that Catholicism never actually stated that men should control women. Relationships between men and women are based on equality and respect for each other as human beings, and these religious values intersect with A's (and Utami's) feminist values. The intersectionality between gender and religion suggests that religion should be considered in areas of feminist research. According to Klassen (2003), to interrogate patriarchal social systems and worldviews, religion must be taken into account. The significant role of religion in women's lives cannot simply be put aside because it is considered to contribute to women's oppression. Religion is also a source of strength and spirit for women facing life difficulties; Klassen argues that religion and spirituality can provide feminists with rich materials for feminist theorising (Klassen 2003). I argue that understanding Utami's rewriting of her critique of marriage makes sense if we consider religion and gender together. To examine the issue of women in developing countries such as Indonesia, for instance, it is impossible to ignore the subject of religion. Indonesia is a large Muslim country where Islamic values frame a cultural context. Further religious conflict and fundamentalist Islamic attacks on non-Muslim (and Muslim) minorities have increased since the end of the New Order. Situated in this context, Utami's political stance in relation to women's oppression has broadened along with a rise in Indonesia of the religious oppression of minority groups. Her involvement with Catholic community activism shows her concerns for religious minority groups and for emerging threats to new freedoms for women in post-Suharto Indonesia. As Day (2006) has stated, democratisation in Indonesia has been accompanied by an Islamic resurgence. For Utami, being a part of her Catholic community no longer conflicts with her feminist politics. Where *The Single Parasite* offered a critique of compulsory heterosexuality early in the post-Suharto period, *Confessions* tells a story of an activist woman negotiating challenges to patriarchal culture from inside a minority religion in the Indonesian context.

**Funding:** This research received no external funding.

**Data Availability Statement:** Not applicable.

**Conflicts of Interest:** The author declares no conflict of interest.

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
