# Peer review of "Compulsory Heterosexuality in Indonesia: A Literary Exploration of the Work of Ayu Utami"

_religions, doi:10.3390/rel13101002_

Round 1

Reviewer 1 Report

The text is generally well written with a good academic style. The choice of Rich’s theory of compulsory heterosexuality as a tool to analyse Utami’s work is however questionable, especially given the analysis which reveals both that Utami is writing as a heterosexual woman who ultimately finds her place within the Catholic minority and the importance of understanding the religious context of the community being studied: Both Catholicism and Islam are faiths that strongly promote heterosexuality. The conclusions also reveal a weakness in the initial contextualisation as this omits any literature relating to the impact of the dominant religion, Islam, on the social and cultural norms of Indonesian society.

Using narrow and extreme methodologies such as that of Rich can not only marginalise the impact of one’s work but may also be accused of ‘erasing’ or at least marginalising the validity of heterosexual women’s sexual experiences which are the norm.

A more robust analysis would have also drawn on the literature which explores women’s sexuality and its expression from within the dominant Muslim community,  to evaluate whether Utami’s writing is limited by her minority perspective.

I would also observe that while it is clear that in Muslim societies women experience much greater societal pressures to marry, the article currently seems to imply that only they that are subject to this when in fact men too are strongly encouraged to marry and within the Islamic texts that are so influential now on Indonesian society homosexuality is discussed and condemned almost exclusively in the context of men.

Author Response

Dear Reviewer,

Thank you for reviewing my article. I have revised it accordingly. There are some points that I would like to highlight:

  • Related  to the the use of Rich's theory, the theory is applied in the context that Indonesia is a country that has the gender ideology of emphasizing  women's roles as wives and mother in the service of her husband and the nation. and only consider women as an adult only if joining the heterosexual marriage therefore, those of outside of heterosexual marriage (single women, unmarried couple, homosexuals) are enforced to conform to it.
  • - While it is true that men and homosexuals also receive the same pressure to marry, the focus of the article is single women in relation to gender ideology of wives and motherhood.  Single men's sexual behaviour outside of marriage is tolerated because leaving no sign, while single women engaged in illicit premarital sex can shamed or even abused and can destroy her reputation and affect her future relationship.

Reviewer 2 Report

This manuscript is quite interesting. The topic is compelling. However, the writing meanders and both the structure and argument of the article need considerable tightening. 

The author repeatedly says what they are going to do in the first half of the article (the first 9 pages) rather than moving their argument forward. So, for example, the paragraph at the end of page 4 says basically the same thing that has been already said on page 2 and at the top of page 4. 

The author makes some statements that are puzzling or just not true, like the one on page 3. “The institution of marriages has historically been the site of multiple interests: to have children, to allow sexual expression, to maintain family values…But none of those interests relate to women’s interests.  This is simply not the case.  Many Indonesian women want children, are interested in sex, and aspire to maintain family values.

The author includes a very long section on "Gender and Sexuality in post-Suharto Indonesia" (pages 5-9) that is rather like an enumeration of gender-related developments in Indonesia without a clear indication of why these are relevant to the author's argument regarding Utami.  The author's strategy of introducing a list of gender related issues in post-Surharto Indonesia and then describing Utami's work doesn't do the work of clearly linking the two together.

On page 9 of 18, or halfway through the article, the author says under a section called "Methods" they will be undertaking a discourse analysis of the two texts in order to understand Utami's shift in perspective (from saying she will never marry to in fact marrying) but the author needs to provide more details from the second novel to explain Utami's change of heart or to set up the shift in a way that the reader can appreciate its implications.  What specifically does A find in Catholicism that convinces her that despite her previous conviction that the religion is patriarchal and biased against women -- as seen for example in the male-only priesthood -- in fact, supports an equal view of partners in marriage?  On page 13 the author writes:

She cannot accept the fact that Catholicism discriminates against women.  And then the next sentence is "She later discovers that Catholicism actually has an egalitarian concept of marriage and becomes determined to marry Rik in the Church." (13)  A rather abrupt turn around! This is the focal point of the article – the change in Utami’s attitude towards marriage -- yet the author does not unpack for the reader sufficient details from the novel to understand why A, after so much time & conviction, suddenly changes her mind.

It becomes clearer in the discussion that follows that this rather abrupt change of mind upsets Utami's readers as well & is a source of considerable debate, but it is not always clear in this section whether the author is referring to Utami's own statements and her own life or the those of the protagonist in her novel (complicated by the fact that the novel is more or less autobiographical), or Utami's statements outside of her writing (mentioned at the bottom of page 13)

In short, the reader has to do too much work to make connections and understand arguments that the author does not clearly draw out. 

Despite these limitations, the article is of considerable interest.  I think with some attention to the presentation of the material and more careful organization and structuring of the author's main argument, the piece could be an important contribution to Indonesian gender & lit studies.

Author Response

Dear Reviewer, 

Thank you for reviewing my paper. I have written the revision. 

  • I have reorganised  the article and provide backgorund to provide the context.
  • The paragraph "The institution of marriage historically has been rewritten to avoid confusion.
  • I have added explanation in order to better explain the part
  • I have rewritten the part explaining the character's change of mind

Reviewer 3 Report

This paper on marriage for women in Indonesia has been submitted to the journal Religions, yet it is not properly framed in the context of religions in Indonesia. Most of the article would seem to be oriented to the field of gender relations or women's studies. There is nothing wrong with that per se, but this is a journal concerned with religions. The author is strongly recommended to revise the paper to include an adequate explanation, up-front, of the landscape of religions and religious tensions in Muslim-majority Indonesia over the past two decades or so. Nancy Smith-Hefner's excellent and comprehensive 2019 book Islamizing Intimacies should be a key point of reference in the revised paper. If there is insufficient framing of the troubled religious context of Indonesia, the reader will struggle to understand the arguments made by the author about feminist writer Ayu Utami's 'shift' on her original expressed choice not to marry. The conclusion of the paper, that Utami's choice to marry after all reflects her position within the besieged minority religion of Catholicism, will not make sense unless the conflictual religious situation in Indonesia is much better explained for the reader. For example, these words at the very end of the paper need to be re-framed and strongly emphasised up-front, at the start, and the context filled out for the readers who are unlikely to know much about Indonesia : 

Further religious conflict and fundamentalist Islamic attacks on non-Muslim (and Muslim) minorities have increased since the end of the New Order. Situated in this context, Utami’s political stance in relation to women’s oppression has broadened along with a rise in oppression of RELIGIOUS mi-nority groups. Her involvement with Catholic community activism shows her concerns for RELIGIOUS minority groups and for emerging threats to new WOMEN'S freedoms in post-Suharto Indonesia (please note my insertions in capital letters.

Round 2

Reviewer 3 Report

The revisions are appropriate and they have much improved the paper